# Targeted gene therapy and cell reprogramming in Fanconi anemia

Paula Rio[1,2,†], Rocio Baños[1,2,†], Angelo Lombardo[3,†], Oscar Quintana-Bustamante[1,2], Lara Alvarez[1,2], Zita Garate[1,2], Pietro Genovese[3], Elena Almarza[1,2], Antonio Valeri[1,2], Begoña Díez[1,2], Susana Navarro[1,2], Yaima Torres[4], Juan P Trujillo[2,5], Rodolfo Murillas[6], Jose C Segovia[1,2], Enrique Samper[4], Jordi Surralles[5], Philip D Gregory[7], Michael C Holmes[7], Luigi Naldini[3,8,**] & Juan A Bueren[1,2,*]

## Abstract

Gene targeting is progressively becoming a realistic therapeutic alternative in clinics. It is unknown, however, whether this technology will be suitable for the treatment of DNA repair deficiency syndromes such as Fanconi anemia (FA), with defects in homology-directed DNA repair. In this study, we used zinc finger nucleases and integrase-defective lentiviral vectors to demonstrate for the first time that *FANCA* can be efficiently and specifically targeted into the *AAVS1* safe harbor locus in fibroblasts from FA-A patients. Strikingly, up to 40% of FA fibroblasts showed gene targeting 42 days after gene editing. Given the low number of hematopoietic precursors in the bone marrow of FA patients, gene-edited FA fibroblasts were then reprogrammed and re-differentiated toward the hematopoietic lineage. Analyses of gene-edited FA-iPSCs confirmed the specific integration of *FANCA* in the *AAVS1* locus in all tested clones. Moreover, the hematopoietic differentiation of these iPSCs efficiently generated disease-free hematopoietic progenitors. Taken together, our results demonstrate for the first time the feasibility of correcting the phenotype of a DNA repair deficiency syndrome using gene-targeting and cell reprogramming strategies.

**Keywords** cell reprogramming; Fanconi anemia; gene-targeting; iPSCs; zinc finger nucleases

**Subject Categories** Genetics, Gene Therapy & Genetic Disease; Haematology; Stem Cells

## Introduction

The progressive development of engineered nucleases has markedly improved the efficacy and specificity of targeted gene therapy, opening new possibilities for the treatment of inherited and acquired diseases in the clinics (Tebas *et al*, 2014). In contrast to conventional gene therapy with integrative vectors, targeted gene therapy enables the insertion of foreign sequences (i.e., therapeutic genes or small oligonucleotides) in specific sites of the cell genome. Thus, depending on the genetic etiology of the disease, the gene-targeting approach may pursue the correction of a specific mutation or, alternatively, the insertion of the therapeutic transgene into safe loci of the genome, often referred to as 'safe harbors' (Naldini, 2011).

In spite of the advances in the field, the question of whether or not targeted gene therapy will be applicable to diseases where homology-directed repair (HDR) is affected has never been explored. Taking into account that Fanconi anemia (FA) proteins participate in HDR (Taniguchi *et al*, 2002; Yamamoto *et al*, 2003; Niedzwiedz *et al*, 2004; Yang *et al*, 2005; Nakanishi *et al*, 2011) and coordinate the action of multiple DNA repair processes, including the action of different nucleases and homologous recombination (see reviews in Kee & D'Andrea, 2010; Kottemann & Smogorzewska, 2013; Moldovan & D'Andrea, 2009), we aimed to investigate for the first time the possibility of conducting a targeted gene therapy strategy in FA cells.

Genetically, FA is a complex disease where mutations in sixteen different genes (*FANC-A, -B, -C, -D1/BRCA2, -D2, -E, -F, -G, -I, -J/BRIP1, -L –M, –N/PALB2, -O/RAD51C; -P/SLX4; -Q/ERCC4/XPF*) have been reported (Bogliolo *et al*, 2013). Among all these genes, mutations in *FANCA* account for about 60% of total FA patients (Casado *et al*, 2007; Auerbach, 2009). Importantly, while few recurrent mutations (i.e., truncation of exon 4 in Spanish gypsies or mutations

1   Division of Hematopoietic Innovative Therapies, CIEMAT/CIBERER, Madrid, Spain
2   Instituto de Investigación Sanitaria Fundación Jiménez Díaz (IIS-FJD, UAM), Madrid, Spain
3   San Raffaele Telethon Institute for Gene Therapy, San Raffaele Scientific Institute, Milan, Italy
4   NIMGenetics SL, Madrid, Spain
5   Universidad Autónoma Barcelona/CIBERER, Barcelona, Spain
6   Division of Epithelial Biomedicine, CIEMAT/CIBERER, Madrid, Spain
7   Sangamo BioSciences Inc., Richmond, CA, USA
8   Vita Salute San Raffaele University, Milan, Italy
    *Corresponding author. Tel: +34 913 466 518; Fax: +34 913 466 484; E-mail: juan.bueren@ciemat.es
    **Corresponding author. Tel: +02 2643 4681; Fax: +02 2643 4621; E-mail: naldini.luigi@hsr.it
    †These authors contributed equally to this work.

in exons 13, 36, and 38) have been observed in FA-A patients, *FANCA* mutations are generally private mutations, which include point mutations, microinsertions, microdeletions, splicing mutations and large intragenic deletions (Castella *et al*, 2011). Thus, considering the large number of genes and mutations that can account for the FA disease, the insertion of a functional FA gene in a 'safe harbor' locus would lead to the generation of a targeted gene addition platform with a broad application in FA, regardless of the complementation group and mutation type of each patient.

Recent studies by our group and others aiming at the identification of 'safe harbor sites' in the human genome have shown robust and stable expression of transgenes integrated in the human *PPP1R12C* gene, a locus also known as *AAVS1*, across different cell types (Smith *et al*, 2008; Lombardo *et al*, 2011). Additionally, no detectable transcriptional perturbations of the *PPP1R12C* and its flanking genes were observed after integration of transgenes in this locus, indicating that *AAVS1* may represent a safe landing path for therapeutic transgene insertion in the human genome (Lombardo *et al*, 2011). These observations, together with the development of artificial zinc finger nucleases (ZFNs) that efficiently and selectively target the *AAVS1* locus, have facilitated gene editing strategies aiming at inserting therapeutic transgenes in this locus, not only in immortalized cell lines but also in several primary human cell types, including induced pluripotent stem cells (hiPSCs; Hockemeyer *et al*, 2009; DeKelver *et al*, 2010; Lombardo *et al*, 2011; Zou *et al*, 2011b; Chang & Bouhassira, 2012).

Because a defective FA pathway not only predisposes FA patients to cancer (Rosenberg *et al*, 2008) but also to the early development of bone marrow failure due to the progressive extinction of the HSCs (Larghero *et al*, 2002; Jacome *et al*, 2006), our final aim in these studies was the generation of gene-edited, disease-free FA-HSCs, obtained from non-hematopoietic tissues of the patient. Thus, in our current studies, we firstly pursued the specific insertion of the therapeutic *FANCA* gene in the *AAVS1* locus of FA-A patients' fibroblasts. Thereafter, gene-edited FA cells were reprogrammed to generate self-renewing disease-free iPSCs and finally re-differentiated toward the hematopoietic lineage, as previously described with FA cells corrected by conventional LV-mediated gene therapy (Raya *et al*, 2009).

Our goal of conducting a combined approach of gene editing and cell reprogramming in FA cells was particularly challenging taking into account the relevance of the FA pathway both in HDR (Taniguchi *et al*, 2002; Yamamoto *et al*, 2003; Niedzwiedz *et al*, 2004; Yang *et al*, 2005; Moldovan & D'Andrea, 2009; Kee & D'Andrea, 2010; Nakanishi *et al*, 2011; Kottemann & Smogorzewska, 2013) and cell reprogramming (Raya *et al*, 2009; Muller *et al*, 2012; Yung *et al*, 2013). In spite of these hurdles, the strong selective growth advantage characteristic of corrected FA cells allowed us to establish a new approach for the efficient generation of FA HPCs harboring specific integrations of the therapeutic *FANCA* gene in a safe harbor locus.

# Results

### Efficient gene-targeting-mediated complementation of fibroblasts from FA-A patients

To promote insertion of a *FANCA* expression cassette into the *AAVS1* locus, an integrase-defective lentiviral vector (IDLV) harboring

the *EGFP* and *FANCA* transgenes flanked by *AAVS1* homology arms (donor IDLV) was generated (Fig 1A top). In this donor IDLV, *FANCA* is under the transcriptional control of the human PGK promoter. In addition, a promoterless *EGFP* cDNA preceded by a splice acceptor (SA) site and a translational self-cleaving 2A sequence was also included upstream of the *FANCA* cassette. Upon targeted-mediated insertion into *AAVS1*, the *EGFP* cassette will be placed under the transcriptional control of the promoter of the ubiquitously expressed *PPP1R12C* gene, thus allowing the FACSorting of gene-targeted cells (Fig 1A). Besides the donor IDLV, an adenoviral vector expressing a ZFN pair (AdV5/35-ZFN), designed to induce a DNA double-strand break in the *AAVS1* locus, was used to enhance the efficiency of gene targeting in this locus (Hockemeyer *et al*, 2009).

To investigate the feasibility of performing gene targeting in FA-A cells, skin fibroblasts from four FA-A patients with different mutations in *FANCA* were transduced either with the donor IDLV alone, or with the donor IDLV and the AdV5/35-ZFNs simultaneously. Fourteen days after transduction, cells were analyzed by flow cytometry to measure the proportion of EGFP$^+$ fibroblasts. While <0.05% of the cells transduced with the donor IDLV alone were positive for EGFP, 0.2–1.1% of FA fibroblasts that had been co-transduced with the donor IDLV and the ZFNs-AdV were EGFP$^+$ (See Fig 1B and representative analyses in Supplementary Fig S1). Strikingly, the percentage of EGFP$^+$ cells markedly increased during the *in vitro* culture of these cells, reaching levels between 5.5 and 13.4% (Fig 1B), showing the proliferation advantage of gene-edited FA-A fibroblasts.

Because the prolonged *in vitro* culture of FA fibroblasts results in increased rates of cell senescence (Muller *et al*, 2012), in a new set of experiments, fibroblasts from three FA patients (FA-52, FA-123 and FA-644) were transduced with an excisable h*TERT*-expressing LV (Salmon *et al*, 2000) prior to performing the gene-targeting procedure. Transduction of FA fibroblasts with hTERT-LVs resulted in a marked increase in telomerase activity (see representative data in Supplementary Fig S2). Significantly, the proportion of EGFP$^+$ cells was markedly increased (3–4-fold) in *hTERT*-transduced versus untransduced FA fibroblasts from FA patients (Fig 1C), indicating that hTERT improved the efficacy of gene targeting in FA-A fibroblasts. Consistent with data obtained with non-immortalized fibroblasts, when immortalized gene-edited FA fibroblasts were maintained in culture, a progressive increase in the proportion of EGFP$^+$ cells was also observed (see data from geFA-52T in Fig 1D). Strikingly, around 40% of treated FA-A fibroblasts were EGFP$^+$ after 42 days in culture in the absence of any selectable drug (Fig 1D).

PCR analyses with two pairs of primers that amplify, respectively, the 5′ and the 3′ integration junctions between the *EGFP/FANCA* cassette and the endogenous *AAVS1* locus evidenced the insertion of the *EGFP/FANCA* cassette into the *AAVS1* locus of sorted EGFP$^+$ geFA-52T fibroblasts (Fig 1E). In these gene-edited FA fibroblasts, the activity of hTERT was also confirmed (Supplementary Fig S2).

To investigate whether the insertion of the therapeutic h*FANCA* cassette in the *AAVS1* locus of FA-A fibroblasts corrected the cellular phenotype of the disease, the functionality of the FA pathway in FA-52T fibroblasts was tested both before (negative control) and after the gene-targeting procedure. As a positive control, healthy

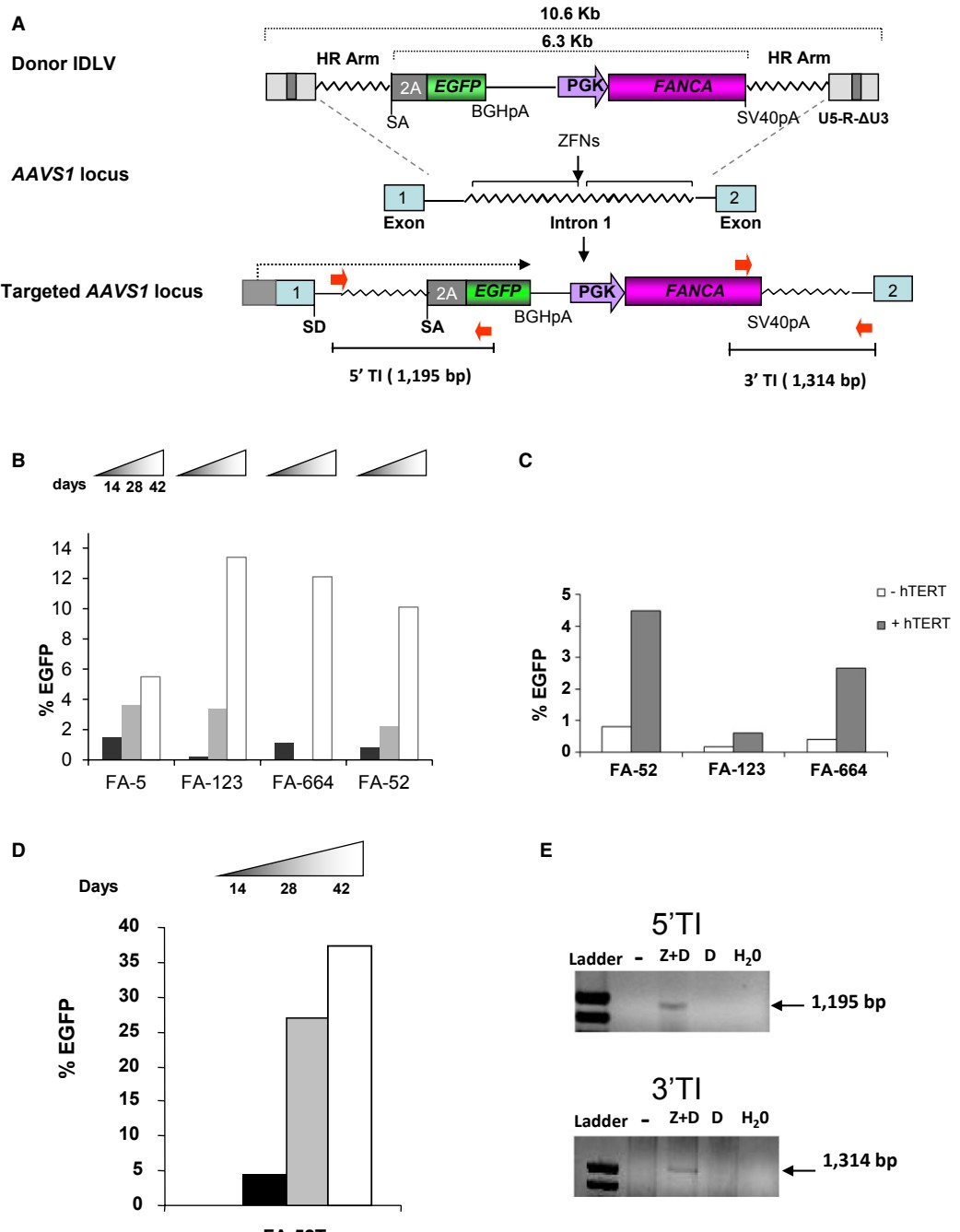

**Figure 1. Efficacy of gene targeting of *FANCA* in the *AAVS1* locus of primary hFA-A fibroblasts.**

A   Top: schematic representation of the donor integrase-defective lentiviral vector (IDLV) used to promote insertion of the *EGFP/FANCA* cassette into the *AAVS1* locus. Middle: *AAVS1* locus with the zinc finger nucleases (ZFNs) target site. Bottom: *AAVS1* locus upon ZFN-mediated targeted insertion of the *EGFP/PGK-FANCA* cassette. Black arrow shows transcription of the *EGFP* from the endogenous *PPP1R12C* promoter. HA, homology arm; SD, splice donor; SA, splice acceptor; BGHpA, bovine growth hormone polyadenylation signal; SV40pA, simian virus 40 polyadenylation signal. Constituents of the LTR (U5-R-ΔU3) are also indicated.

B   Proliferation advantage of targeted Fanconi anemia (FA) fibroblasts (EGFP+ cells) during *in vitro* incubation.

C   Comparative analysis of gene targeting in FA-A fibroblasts, untransduced or transduced with a lentiviral vector expressing h*TERT*. Analyses were performed 14 days after gene targeting.

D   *In vitro* proliferation advantage of targeted FA fibroblasts (EGFP+) previously transduced with hTERT (FA-52T fibroblasts).

E   Targeted integration analysis of the *EGFP/PGK-FANCA* cassette into the *AAVS1* site by PCR using primers specific for the 5′ or 3′ integration junctions (red arrows in the top schematic) defined as 5′ TI or 3′ TI, respectively.

    

donor fibroblasts (H.D. Fib) were analyzed in parallel. The presence of nuclear FANCD2 foci, fully dependent on the expression of all the FA core complex proteins, including FANCA (Garcia-Higuera *et al*, 2001), was determined in these samples after DNA damage induced by mitomycin C (MMC). In contrast to uncorrected FA-52T fibroblasts (FA-52T Fib.), which did not generate FANCD2 foci even after MMC exposure, a significant proportion of the geFA-52T fibroblasts generated FANCD2 foci, mainly after treatment with MMC, thus

mimicking the response of H.D. fibroblasts (Fig 2A). Because the main characteristic of FA cells is the increased chromosomal instability upon exposure to DNA inter-strand cross-linking (ICL) drugs, we also investigated the response of both uncorrected and gene-edited FA-A fibroblasts to diepoxybutane (DEB). While in FA-52T fibroblasts DEB induced a significant increase in the number of chromosomal aberrations per cell (from $0.05 \pm 0.05$ to $1.7 \pm 0.46$ aberrations/cell)— including chromatid breaks and

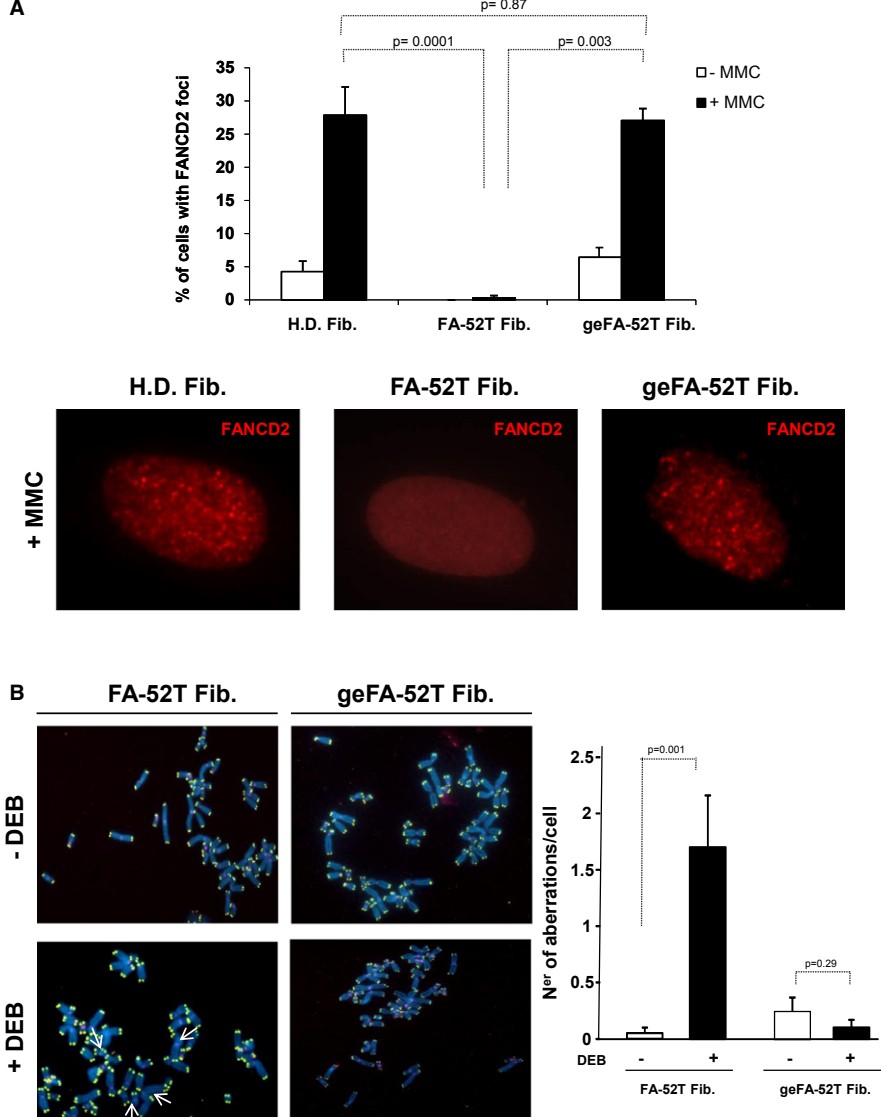

**Figure 2. Phenotypic correction of the gene-edited FA-A fibroblasts.**

A   Top: histogram showing the percentage of FA-A fibroblasts, unstransduced or co-transduced with the donor integrase-defective lentiviral vector (IDLV) and the AdV5/35-ZFNs (geFA-52T Fib), showing FANCD2 foci in the absence or the presence of mitomycin C (MMC). Bottom: representative images of FANCD2 foci (red) in cells shown in the top histogram, after MMC treatment.

B   Chromosomal instability induced by diepoxybutane (DEB) in untreated (FA-52T) and gene-edited FA fibroblasts (geFA-52T Fib). Left: representative FISH analysis was performed by staining telomeres (in green), centromeres (in pink) and chromosomes (in blue). Right: histogram showing the number of chromosomal aberrations per cell.

Data information: Values are shown as mean $\pm$ s.e. from three independent experiments (A) or analysis of twenty different metaphases per group (B). All *P*-values were calculated using two-tailed unpaired Student's *t*-test.

    

radial chromosomes, typically found in FA patients' cells—the same DEB treatment did not induce any increase in the number of chromosomal aberrations in geFA-52T fibroblasts (Fig 2B).

Taken together, these results show the feasibility of correcting the phenotype of FA cells using gene targeting strategies, in particular by promoting the insertion and expression of *FANCA* in the *AAVS1* safe harbor locus of fibroblasts from FA-A patients.

## Efficient generation of disease-free iPSCs from FA fibroblasts corrected by gene targeting

To generate disease-free FA-iPSCs, FA fibroblasts subjected to gene editing (geFA-123, geFA-52 and geFA-52T) were first enriched for EGFP$^+$ cells by cell sorting and then reprogrammed using a polycistronic excisable LV expressing the human *SOX2, OCT4, KLF4,* and *cMYC* transgenes from the *EF1A* promoter (STEMCCA vector; Somers *et al*, 2010). Consistent with previous observations (Raya *et al*, 2009), uncorrected FA fibroblasts did not generate iPSCs after reprogramming, even after transduction with the TERT-LV (data not shown). Although several iPSC-like colonies were generated from gene-edited FA-123 fibroblasts (115 AP$^+$ cells/100,000 fibroblasts), no stable iPSC lines could be generated from FA fibroblasts simply subjected to gene editing, most probably because of the pro-senescence nature of these cells. In marked contrast to these observations, the reprogramming of FA fibroblasts that were first transduced with the hTERT-LV and then subjected to gene editing generated 230 iPSC-like clones, most of which could be maintained after serial *in vitro* passages (Supplementary Fig S3). Twelve iPSC clones generated from geFA-52T fibroblasts were further expanded and differentiated into fibroblasts to perform additional studies to confirm the integration site of the EGFP/*FANCA* construct. First, qPCR analyses were conducted to determine the mean copy number per cell of the *EGFP/FANCA* cassette. As shown in Supplementary Table S1, 11 out of the 12 geFA-iPSC clones analyzed were positive for *EGFP* integration and contained an average of $0.98 \pm 0.44$ *EGFP* copies per cell. The only iPSC clone that did not harbor any EGFP copy (clone 5) did not progress more than six passages in culture.

To investigate whether the *EGFP/FANCA* cassette was specifically integrated in the *AAVS1* locus of all these iPSC clones, 3′ primers previously used in analyses of Fig 1E were used. As shown in Supplementary Table S1, all iPSC clones that were positive for integration of the cassette were also positive for the PCR band corresponding to the specific insertion in the *AAVS1* locus.

Three geFA-iPSC clones (clones 16, 26 and 31) were selected for further characterization. The pluripotency of these gene-corrected clones was first analyzed both by alkaline phosphatase (AP) staining and immunohistochemistry staining of different pluripotency genes. Representative pictures in Fig 3A and Supplementary Fig S4A showed that all tested geFA-iPSCs clones were highly positive for AP, NANOG, TRA-1-60, OCT4, and SSEA-4 expression. RT-qPCR analyses of the expression of endogenous pluripotency genes *NANOG, OCT4, SOX2, KLF4,* and *cMYC* were consistent with the pluripotent nature of these clones (Supplementary Fig S4B). In all cases, a very low expression of the ectopic reprogramming transgenes was found, indicating substantial inactivation of the *EF1A* promoter present in the reprogramming vector. As expected for *bona fide* iPSC clones, OCT4 and NANOG

promoters were hypomethylated in gene-corrected FA-iPSC clones, in clear contrast to the high level of methylation observed in H.D. fibroblasts (Supplementary Fig S4C). To further demonstrate the pluripotency of geFA-iPSC16 cells *in vivo*, cells were subcutaneously inoculated in NSG mice. Characteristic teratomas containing complex structures representing the three embryonic germ layers were observed 8–10 weeks after implantation. Immunofluorescence staining confirmed the expression of definitive endoderm markers (Fox2A), neural structures that expressed neuroectodermal markers (ß-III-tubulin) and the generation of mesoderm (Brachyury) and mesoderm derivatives tissue such as muscle (α-SMA; Fig 3B).

To confirm the insertion of the FANCA cassette into the *AAVS1* locus in the gene-corrected FA-iPSC clones, Southern blot analyses were performed on genomic DNA extracted from gene-edited geFA-iPSC clones 16, 26, and 31. Blots hybridized with probes for the exogenous *EGFP* and the endogenous *AAVS1* genes confirmed the monoallelic integration of the *EGFP/FANCA* cassette into the *AAVS1* locus and the absence of random integration in any of the three tested clones (Fig 3C,D).

Once demonstrated the generation of *bona fide* gene-edited FA-iPSCs, in the next set of experiments, we aimed to verify whether these geFA-iPSCs were disease free, as shown for their parental gene-edited FA fibroblasts (Fig 2). First, we verified by qRT-PCR that h*FANCA* mRNA levels corresponding to the three tested geFA-iPSC clones were very similar to levels observed in the control ES cell line and markedly higher when compared to uncorrected FA-52T fibroblasts (Fig 4A). Western blot analysis confirmed the expression of FANCA in all the three tested clones (Fig 4B). Even more, since FANCA is necessary for the relocation of FANCD2 to damaged DNA sites, we investigated the presence of nuclear FANCD2 foci in three geFA-iPSC clones exposed to MMC. As shown in Fig 4C, these analyses further confirmed the expression and functionality of FANCA in the three tested geFA-iPSC clones. Consistent with the restored FA pathway of gene-edited FA-iPSCs, DEB did not induce a significant increase in the number of chromosomal aberrations in FA-corrected cells. Remarkably, the number of chromosomal aberrations in geFA-iPSCs ($0.2 \pm 0.1$ aberrations/cell; Fig 4D) was ten times lower to the number observed in their parental uncorrected fibroblasts (see Fig 2B).

To assure the identity of the different geFA-iPSC clones, the presence of the original pathogenic mutations described in patient FA-52 (c.710-5T>C and c.3558insG) was investigated by Sanger sequencing both on FA-52T fibroblasts and geFA-iPSC clones 16, 26, and 31 (Supplementary Fig S5). The confirmation of both pathogenic mutations in the three tested geFA-iPSCs, together with our observations showing that all stable iPSC clones contained the *AAVS1*-targeted *FANCA* gene (Supplementary Table S1) and had a functional FA pathway, demonstrates that the disease-free nature of gene-edited FA-iPSCs is a consequence of the functional insertion of *FANCA* within the *AAVS1* safe harbor site of these reprogrammed FA cells.

Aiming to excise the STEMCCA vector from the genome of geFA-iPSCs, cells from clone 16 were transduced with an IDLV co-expressing the Cre recombinase and the Cherry fluorescence marker (Papapetrou *et al*, 2011). Thereafter, individual colonies were isolated to select those clones with a lower number of copies of the STEMCCA provirus. Two clones were selected: Excised clones

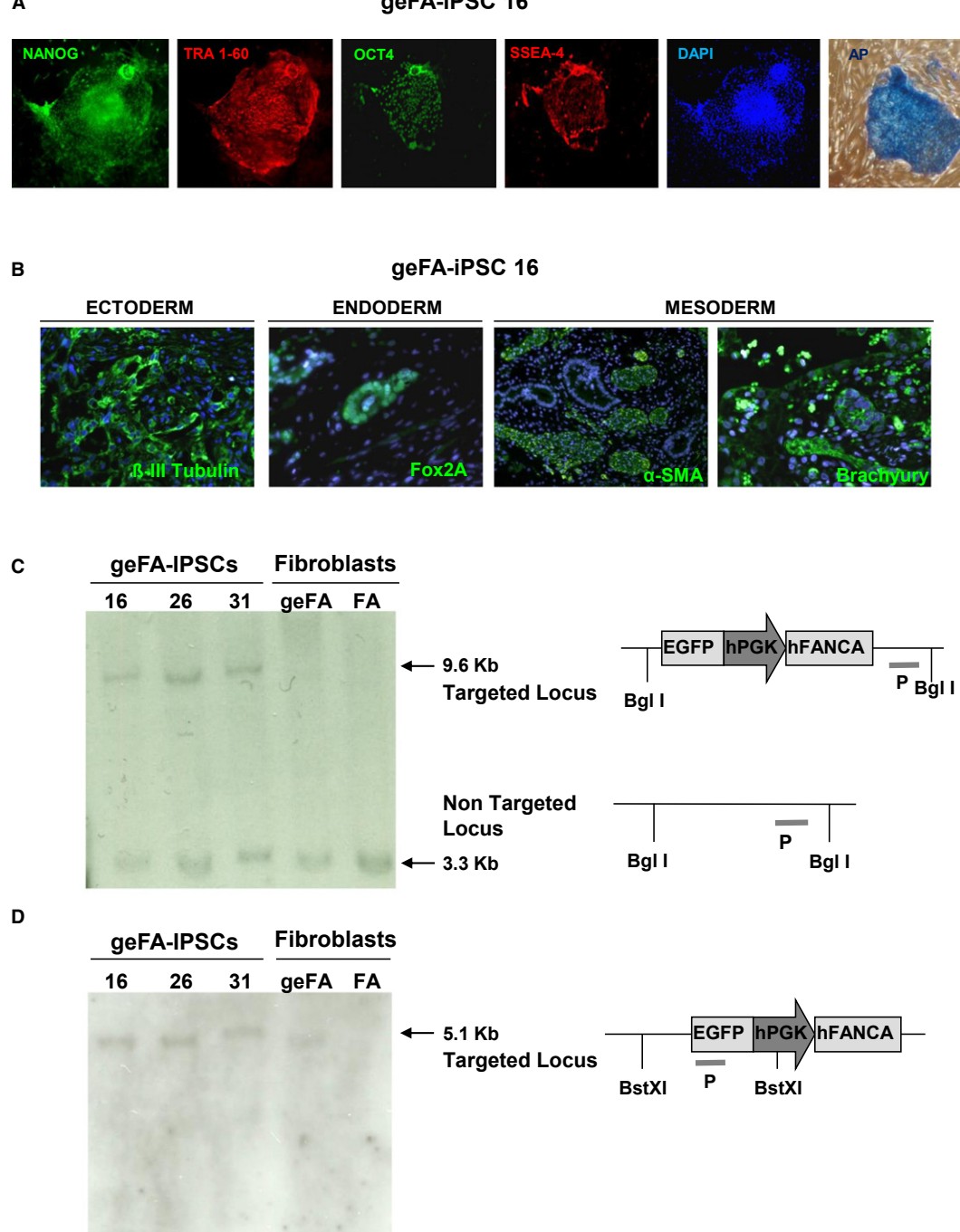

**Figure 3.  Pluripotency characterization and insertion site analyses of gene-edited FA-A iPSCs.**

A   Expression of TRA1-60, SSEA-4, OCT4, and NANOG pluripotency markers by immunofluorescence staining of gene-edited FA-iPSCs (geFA-iPSCs; clone 16).

B   Immunofluorescence analysis of ectoderm (β-II-tubulin), endoderm (Fox2A), and mesoderm (α-SMA and Brachyury) in teratomas generated from geFA-iPSCs (clone 16).

C   Southern blot analysis of genomic DNA extracted from the indicated gene-corrected FA iPSC clones (geFA-IPSCs) and from parental fibroblasts, either unmanipulated (FA) or after gene editing (ge-FA iPSCs, clones 16, 26 and 31). Genomic DNA was digested with BglI and hybridized with a probe for *PPP1R12C*. The band of 9.6 kb corresponds to the targeted integration in *PPP1R12C*, while the 3.3 kb correspond to the untargeted allele.

D   Southern blot analysis of samples shown in (C) digested with BstXI and hybridized with a probe (P) for *EGFP*. One single band of 5.1 kb is expected for specific integrations in *PPP1R12C*.

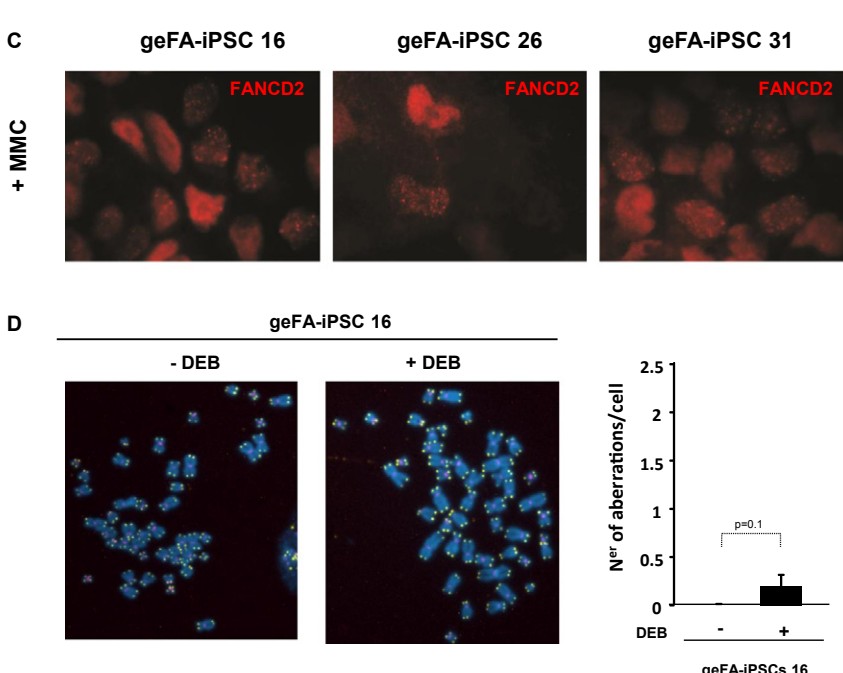

**Figure 4. Disease-free Fanconi anemia phenotype of corrected geFA-iPSCs.**

A  Histogram showing the levels of *hFANCA* expression in gene-edited FA-iPSC clones and human ES (H9) relative to untreated FA-52T fibroblasts. Data are shown as mean ± s.e. of three different analyses.

B  Western blot analysis showing FANCA expression in geFA-iPSC clones in comparison with fibroblasts from HD and a FA-A patient.

C  Representative immunofluorescence analysis of FANCD2 foci in geFA-iPSCs after DNA damage with mitomycin C (MMC).

D  Chromosomal instability induced by diepoxybutane (DEB) was also tested in geFA-iPSC 16. FISH analysis was performed using probes to detect telomeres (green), centromeres (pink) and chromosomes (blue). Right: histogram showing the number of chromosomal aberrations per cell.

Data information: Data are shown as mean ± s.e. from three different experiments (A) or analysis of twenty different metaphases per group (D). All *P*-values were calculated using two-tailed unpaired Student's *t*-test.

16.1 and 16.2, with a number of 0.35 ± 0.10 and <0.05 copies/cell, respectively. In clone 16.2, the excision of the h*TERT* provirus was also confirmed (<0.05 copies as deduced from q-PCR analyses). RT-qPCR analysis performed in these two subclones showed the persistent expression of endogenous pluripotency genes (*SOX2,*

*OCT4, KLF4, NANOG, and cMYC*) and the absence of ectopic transgenes expression (Supplementary Fig S6A). As expected from *bona fide* pluripotent iPSC clones, these two clones generated teratomas with structures characteristics of the three germ layers (Supplementary Fig S6B).

## Analysis of the genetic stability of gene-edited FA fibroblasts and iPSCs

Because of the chromosomal instability of FA cells, we investigated by means of karyotype analyses and aCGH analyses whether the different manipulations of FA-52 fibroblasts and their corresponding iPSCs induced chromosomal instability. As shown in Table 1, no evident karyotype or aCGH abnormalities were observed in expanded FA-52 parental fibroblasts when compared with a reference human DNA sample. Even more, the transduction with *hTERT-LV* and the gene-editing process did not induce evident chromosomal abnormalities in these cells. Reprogrammed geFA-52 iPSCs also had a normal karyotype, although a deletion in the 16p12.2p12.1 locus was noted in the aCGH analysis. After excision with the Cre recombinase, in addition to the 16p deletion, a mosaic trisomy in chromosome 5 was observed (See Table 1 and Supplementary Fig S7).

## Generation of disease-free hematopoietic progenitors from gene-edited FA-A iPSCs

In experiments corresponding to Fig 5 and Supplementary Figs S8 and S9, we investigated whether hematopoietic progenitor cells derived from gene-edited FA-iPSCs were disease-free. To conduct these experiments, embryoid bodies from geFA-iPSCs were incubated with hematopoietic cytokines as described in Materials and methods. As shown in representative analyses from Supplementary Fig S8A, the hematopoietic differentiation of geFA-iPSCs after 21 days of *in vitro* stimulation was demonstrated by the presence of hematopoietic precursors (CD43$^+$/CD34$^+$), committed hematopoietic progenitors (CD34$^+$/CD45$^+$) and also mature hematopoietic cells (CD34$^-$/CD45$^+$). When the hematopoietic differentiation of excised and non-excised iPSC clones was compared, the proportion of CD45$^+$ and CD34$^+$/CD45$^+$ was consistently increased in the case

of the excised *vs* the non-excised clones (see data from two independent experiments in Fig 5A and Supplementary Fig S8). Consistent with the flow cytometry data, granulo-macrophage and erythroid colonies were generated by geFA-iPSC-differentiated cells in methylcellulose. As it was observed in the flow cytometry studies, higher numbers of hematopoietic progenitors were generated by excised versus non-excised geFA-iPSC (Fig 5B). In all instances, colonies derived from geFA-iPSC were almost as resistant to MMC as healthy cord blood progenitor cells, in contrast to the MMC hypersensitivity observed in BM progenitors from FA patients (Fig 5C).

Finally, to investigate whether gene-edited FA-iPSCs were also able to differentiate toward the hematopoietic lineage *in vivo*, one of the teratomas generated by the excised geFA-52 iPSCs (clone 16.2) was analyzed for the presence of human hematopoietic markers. As shown in Supplementary Fig S9, 3% of the cells present in this teratoma consisted on hCD45$^+$/mCD45$^-$ cells. Within this population, 3.5% corresponded to hCD34$^+$ cells, thus revealing the *in vivo* differentiation potential of this clone.

# Discussion

Thanks to the development of artificial nucleases capable of generating DNA double-strand breaks (DSBs) in pre-determined sequences of the genome (Porteus & Baltimore, 2003; Urnov *et al*, 2010; Cong *et al*, 2013; Joung & Sander, 2013), targeted gene therapy is entering into the clinics (Tebas *et al*, 2014). Whether these approaches will be amenable to the treatment of DNA repair deficiency syndromes such as FA is, however, uncertain. In this respect, it is currently known that FA proteins participate in maintaining the genomic stability of the cell and coordinate the actions of multiple repair processes, including HDR (Kottemann & Smogorzewska, 2013), making these cells particularly appropriate for investigating the feasibility of performing targeted gene therapy in syndromes associated with DNA repair defects and genome instability. Although the mechanisms explaining how the FA pathway promotes HDR are still unclear, most evidence suggests that the monoubiquitination of FANCD2—which is critically dependent on the presence of all the FA core complex proteins, including FANCA—is essential for the recruitment of several HDR factors (such as BRCA1, BRCA2, and RAD51) to damaged chromatin (see review in Kee & D'Andrea, 2010).

To investigate whether gene targeting was feasible in FA cells we focused on the most frequent FA complementation group, FA-A (Casado *et al*, 2007; Auerbach, 2009), and investigated the possibility of inserting the therapeutic transgene in a safe harbor locus of the human genome—the *AAVS1* locus (Lombardo *et al*, 2011).

Strikingly, our first results in Fig 1 clearly demonstrate the feasibility of performing gene targeting in FA-A cells with significant efficacies (up to 4%), comparable with efficacies reported in primary cells competent for DNA repair (DeKelver *et al*, 2010; Lombardo *et al*, 2011; Sebastiano *et al*, 2011; Soldner *et al*, 2011; Zou *et al*, 2011a). The feasibility of performing gene targeting in FA-A cells could be explained by different hypotheses. First, as previously described in other systems (Matrai *et al*, 2011; Peluffo *et al*, 2013), a transient though early expression of FANCA may be induced by the donor IDLV, thus facilitating the insertion of the exogenous therapeutic cassette through a HDR process. Besides

**Table 1.  aCGH analysis in FA-52 fibroblasts prior to and after gene editing and in gene-edited IPSCs-derived clones**

| Cells | aCGH result | | | Karyotype |
| | Alteration | Locus | OMIM GENES | |
|---|---|---|---|---|
| FA-52 fibroblasts[a] | – | – | – | 46 XY |
| geFA-52T fibr.[b] | – | – | – | 46 XY |
| geFA-52T iPSC clones | | | | |
| Clone 16[c] | Deletion | 16p12.2p12.1 | * | 46 XY |
| Clone 16 Ex[c] | Deletion | 16p12.2p12.1 | * | 46 XY |
| | Mosaic trisomy | 5 | – | 46 XY |

*EEF2K, CDR2, HS3ST2, SCNN1G, SCNN1B, COG7, GGA2, EARS2, NDUFAB1, PALB2, DCTN5, PLK1, ERN2, PRKCB, CACNG3, RBBP6.

[a]Comparison analyses between expanded fibroblasts from patient FA-52 (FA-52 fibroblasts) and a reference male DNA sample.

[b]Comparison analyses between expanded, TERT-transduced, and gene-edited FA-52 fibroblasts (geFA-52T fibr.) with respect to FA-52 fibroblasts.

[c]Comparison analyses between geFA-52T iPSCs clone 16 and clone 16 Ex (after excision of the reprogramming cassette) and FA-52 fibroblasts.

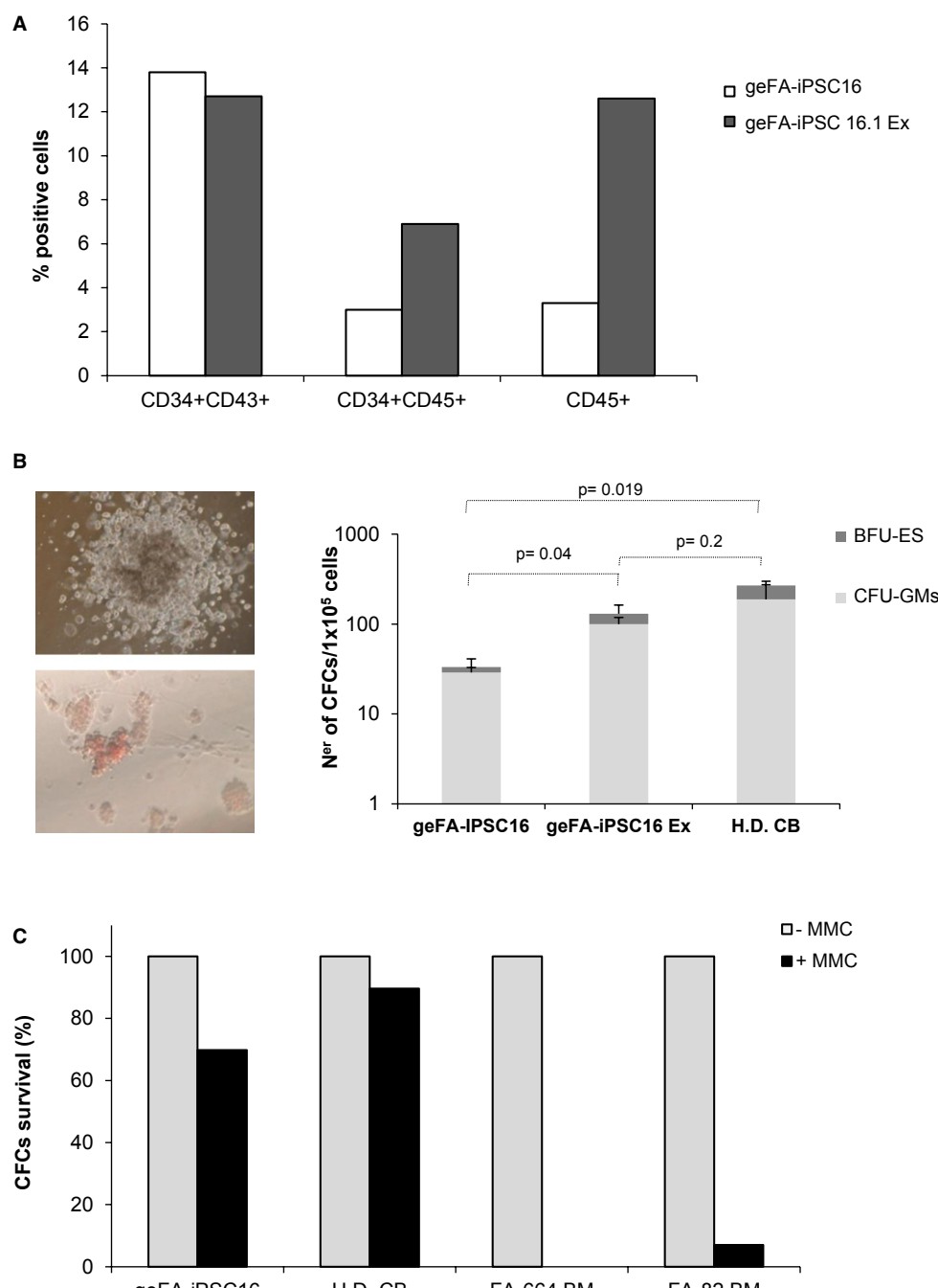

**Figure 5. Hematopoietic differentiation of gene-edited FA-IPSCs.**

A   Analysis of the percentage of CD43$^+$CD34$^+$, CD45$^+$CD34$^+$, and CD45$^+$ cells generated by unexcised and excised geFA-iPSCs (clones 16 and Ex 16.1).

B   Left: Representative pictures of hematopoietic colonies generated by geFA-iPSCs. Right: Analysis of the clonogenic potential of unexcised and excised ge-FAiPSCs (clones 16 and Ex 16.1) in comparison with H.D. cord blood cells.

C   Survival to mitomycin C (MMC) of CFCs obtained from geFA-IPSCs (clone 16) in comparison with BM CFCs from two different FA patients (FA-664 BM and FA-82 BM) and with CFCs from a healthy cord blood (H.D. CB).

Data information: Values are shown as mean ± s.e. of three experiments. All *P*-values were calculated using two-tailed unpaired Student's *t*-test.

this hypothesis, we should contemplate the possibility that the limited HDR activity of FA-A cells (Nakanishi *et al*, 2005, 2011) could be sufficient to facilitate the ZFN-mediated integration of our donor IDLV in the *AAVS1* site. Finally, although the integra-

tion of the therapeutic cassette in the *AAVS1* locus might have occurred through an HDR-independent process, as reported in other models (Anguela *et al*, 2014), PCR and Southern blot analyses showed the expected amplicons and band length for targeted

integration of the cassette, strongly suggesting that *AAVS1* targeting took place through a HDR mechanism. In this respect, while the specificity of gene targeting might be reduced in FA cells, our data clearly show that all the FA-iPSC clones harbored one single copy of *FANCA* specifically integrated in the *PPPR12C* target gene (Table 1). Consequently, this result further supports the efficacy and the specificity of our gene targeting approach.

With the main objective of preventing the predisposition to senescence of FA cells (Muller *et al*, 2012), the transduction of hTERT-LV in FA-A fibroblasts induced an unexpected effect in these cells, which consisted of a significant increase in the efficacy of gene editing (Fig 1). Whether or not this effect is specific for FA cells or whether it is simply mediated by the enhanced proliferation rate of TERT-transduced FA cells is currently unknown. Nevertheless, to the best of our knowledge, the improved gene targeting mediated by hTERT observed in our experiments constitutes a new finding that has not been previously reported in any other experimental model. The observation that transduction with hTERT also facilitates the generation of gene-edited FA-iPSCs is consistent with previous data showing the relevance of hTERT in cell reprogramming (Batista *et al*, 2011; Pomp *et al*, 2011; Winkler *et al*, 2013). In safety terms, even though the *hTERT* provirus could be efficiently excised from transduced cells with the Cre recombinase, further approaches based on the transient expression of hTERT during gene editing and/or cell reprogramming would constitute safer approaches to limit potential genomic insults during the *ex vivo* manipulation of the samples.

Interestingly, EGFP analyses in gene-edited FA fibroblasts showed that in the absence of any artificial selection process, a progressive increase in the proportion of targeted cells (up to 40% after 42 days in culture) was observed, mimicking the improved growth proliferation properties of FA precursor cells in mosaic patients (Waisfisz *et al*, 1999; Gregory *et al*, 2001; Gross *et al*, 2002) or in experimental models of FA gene therapy (Rio *et al*, 2008). Consistent with previous observations in FA cells corrected by LV-mediated gene therapy (Raya *et al*, 2009), this proliferation competence of FA-corrected cells was particularly remarkable when samples were subjected to cell reprogramming, confirming the relevance of the FA pathway during the process of iPSC generation. Similar conclusions were obtained in two additional studies (Muller *et al*, 2012; Yung *et al*, 2013), although these studies showed that reprogramming of FA cells can occur, albeit with a very low efficiency compared to gene-complemented FA cells.

Studies in Figs 2 and 4 showing the generation of nuclear FANCD2 foci and the chromosomal stability of gene-edited FA fibroblasts and iPSCs upon exposure to ICL drugs demonstrate that the specific targeting of *FANCA* in the *AAVS1* locus has completely corrected the phenotype of FA-A fibroblasts and *bona fide* iPSCs. Although transduction of FA fibroblasts with the hTERT-LV might have had consequences upon the genetic instability of FA cells, our karyotype and aCGH studies indicate that neither the expansion nor the transduction with hTERT-LV or the gene-editing processes induced evident chromosomal abnormalities in FA fibroblasts. In contrast to these results, data in Table 1 and Supplementary Fig S7 showed the presence of chromosomal abnormalities in reprogrammed and excised geFA-iPSCs. Importantly, different genetic defects have also been reported in non-FA-iPSCs (Mayshar *et al*, 2010; Gore *et al*, 2011; Laurent *et al*,

2011; Cheng *et al*, 2012; Ruiz *et al*, 2013) that were associated with the generation of the iPSCs (Mayshar *et al*, 2010; Gore *et al*, 2011; Hussein *et al*, 2011; Laurent *et al*, 2011) and/or with mutations that pre-existed in the somatic population of origin (Young *et al*, 2012). This indicates that the presence of chromosomal abnormalities in our iPSCs is not exclusive of their FA genetic background and that the different mechanisms accounting for mutations in non-FA-iPSCs would be applicable to our geFA-iPSCs.

Consistent with the previous study showing the generation of disease-free FA-iPSCs through conventional gene therapy approaches (Raya *et al*, 2009; Muller *et al*, 2012), our new study shows the efficient hematopoietic differentiation of gene-edited FA-iPSCs. Moreover in the current study, we observed the generation of increased numbers of hematopoietic progenitors from geFA-iPSCs subjected to excision of the reprogramming cassette, confirming previous observations showing that the residual expression of reprogramming genes limits the iPSC differentiation potential (Ramos-Mejia *et al*, 2012). The hematopoietic differentiation observed in these experiments and the robust expression of *FANCA* targeted into the safe harbor *AAVS1* locus should account for the generation of a high number of hematopoietic progenitors with normalized response to MMC.

In summary, our study demonstrates for the first time the possibility of conducting efficient and precise targeted-mediated gene therapy in HDR-deficient cells. Moreover, we show the feasibility of reprogramming these cells to generate iPSC-derived gene-edited hematopoietic progenitors characterized by a disease-free phenotype. Our approach thus constitutes a new proof-of-concept with a potential future clinical impact to optimize the generation of gene-corrected HSCs from non-hematopoietic tissues of patients with inherited diseases, including DNA repair deficiency and genetic instability syndromes, like FA.

# Materials and Methods

### Cell lines and primary fibroblasts from FA-A patients

293T and HT1080 cells (ATCC: CRL-11268 and ATCC: CCL-121) were used for the production and titration of the LVs, respectively. Cells were grown in Dulbecco's modified medium GlutaMAX™ (DMEM; Gibco) supplemented with 10% fetal bovine serum (FBS, Biowhitaker) and 0.5% penicillin/streptomycin solution (Gibco). Skin fibroblasts were obtained from FA-5, FA-123, FA-664, and FA-52 patients and were maintained in DMEM (Invitrogen) supplemented with 20% FBS (Biowhitaker) and 1% penicillin/streptomycin solution (Gibco) at 37°C under hypoxic conditions (5% of $O_2$) and 5% of $CO_2$. Patients were classified as FA-A patients as previously described (Casado *et al*, 2007). The ES4 and H9 (NIH Human Embryonic Stem Cell Registry, http://stemcells.nih.gov/research/registry/) lines of hES cells were maintained as originally described (Raya *et al*, 2008). FA patients and healthy donors were encoded to protect their confidentiality, and informed consents were obtained in all cases according to Institutional regulations of the CIEMAT. All studies conformed the principles set out in the World Medical Association Declaration of Helsinki.

## Vectors

pCCL.sin.cPPT.AAVS1.loxP.SA.2A.GFP.pA.loxP.PGK.FANCA.pA.Wpre donor transfer LV (donor IDLV) was generated using elements from the backbones pCCL.PGK.FANCA.Wpre* (Gonzalez-Murillo *et al*, 2010) and pCCLsin.cPPT.AAVS1.2A.GFP.pA (Lombardo *et al*, 2011). The integrase-defective third-generation packaging plasmid pMD.Lg/pRRE.D64Vint was used to produce IDLV particles (Lombardo *et al*, 2007). pLM.CMV.Cherry.2A.Cre (Papapetrou *et al*, 2011) and pLox.TERT.ires.TK vectors (Salmon *et al*, 2000) were provided by Addgene. For reprogramming experiments, the EF1α STEMCCA lentiviral vector kindly provided by Dr Mostoslavsky was used (Sommer *et al*, 2010). This vector contains the cDNAs for *OCT4, SOX2, c-MYC,* and *KLF4* flanked by loxP sequences for their subsequent excision. ZFNs targeting intron 1 of the *PPP1R12C* gene were expressed from an Adenoviral Vector (AdV5/35) under the control of the CMV promoter (Lombardo *et al*, 2011).

## Cell transduction

For gene editing experiments, fibroblasts from FA-A patients were transduced either with donor IDLV alone (150 ng HIV Gag p24/ml) or together with AdV5/35-ZFNs (multiplicity of infection (MOI) 200). Fourteen days post-transduction, the proportion of EGFP$^+$ cells was determined by flow cytometry (BD LSRFortessa cell analyzer, Becton Dickinson Pharmingen). To immortalize fibroblasts from FA-52 to FA-123 patients, $10^5$ cells were transduced at MOI 1 with the pLox.TERT.ires.TK LV (Salmon *et al*, 2000) for 24 h. To excise the reprogramming cassette and hTERT from established hiPSCs, single cell suspensions were generated by incubation with accutase (Gibco) and transduced for 10 h with the IDLV pLM.CMV.Cherry.2A.Cre. Immediately after transduction, $2 \times 10^4$ cells/10 cm$^2$ dish, expressing Cherry protein, were sorted and new subclones of the parental geFA-IPSCs were generated.

## Hematopoietic differentiation

iPSC colonies were detached using colagenase type IV (Gibco) for 30 min at 37°C, washed and centrifuged at 200× *g*, resuspended in differentiation media composed by KO-DMEM (Gibco) supplemented with 20% non-heat-inactivated FBS (Biowhitaker), 1% NEAA (Lonza; Biowhitaker), L-Glu (1 mM; Invitrogen), β-mercaptoethanol (0.1 mM; Gibco) and hrBMP4 (0.5 ng/ml; Prepotech) and plated in ultra-low attachment plates (Costar). After 2 days, media were replaced by Stempro 34 (Invitrogen) supplemented with 0.5% pen/streptomicin, L-Glu (2 mM; Invitrogen), MTG (40 mM; Sigma), ascorbic acid (50 μg/ml; Invitrogen), hrSCF, hrFlt3 ligand and TPO (100 ng/ml; EuroBioSciences), hrIL3 (10 ng/ml; Biosource), hrIL6 (10 ng/ml; Prepotech), hrBMP4 (50 ng/ml; Prepotech), Wnt11 (200 ng/ml; R&D), and rhVEGF (5 ng/ml; Prepotech). Media were changed every 3–4 days. At day 7, media were replaced by fresh media where rhWnt-11 was substituted by rhWnt-3a (200 ng/ml; R&D). Media were changed every 3–4 days. At day 14 and 21, immunophenotypic analysis of the differentiated cells was performed by flow cytometry, and colony-forming unit assays were conducted (See Supplementary Methods).

## Flow cytometry

Transduction with the AdV5/35-ZFNs and the donor IDLV, was analyzed by flow cytometry analysis (FACSCalibur; Becton Dickinson Pharmingen). Immunophenotypic analysis of the hematopoietic differentiated cells was performed using the following antibodies according to the manufacturer's instructions: phycoerythrin (PE)-Cy7-conjugated anti-human CD34 (BD Pharmingen), PE-conjugated anti-human CD31 (eBiosciences), allophycocyanin (APC)-conjugated anti-human CD45 (BD), and fluorescein isothiocyanate (FITC)-conjugated anti-human CD43 (BD). Fluorochrome-matched isotypes were used as controls. 4′,6-Diamidino-2-phenylindole (DAPI; Roche)-positive cells were excluded from the analysis. Analysis was performed using FlowJo software.

## Inmunofluorescence and Western blot of Fanconi anemia proteins

Analyses of FANCD2 foci were performed by immunofluorescence of primary fibroblasts or iPSCs treated for 16 h with 200 nM of MMC. After MMC treatment, cells were stained with rabbit polyclonal anti-FANCD2 (Abcam, ab2187-50) as previously described (Hotta & Ellis, 2008; Raya *et al*, 2009). Cells with more than ten foci were scored as positive. FANCA expression was analyzed by Western blot (Raya *et al*, 2009) using the following antibodies: hFANCA (ab5063 Abcam) and anti-beta Actin to mouse antibody (ab6276, Abcam) as control. Goat polyclonal antibody to rabbit IgG (HRP; ab6721-1; Abcam) and sheep polyclonal antibody to mouse IgG—H&L (HRP; ab 6808, Abcam) were used as secondary antibodies. Protein quantification was done with Image J software.

## *FANCA* expression by qRT-PCR

The expression of human *FANCA* mRNA was analyzed in the different clones of geFA-iPSCs by real-time quantitative reverse transcriptase-polymerase chain reaction (qRT-PCR; Gonzalez-Murillo *et al*, 2010) using primers described in Supplementary Methods. Parental fibroblasts from FA-52 and ES H9 were used as controls.

## Gene targeting analysis: PCR and Southern blots

For PCR analysis, genomic DNA was extracted with DNeasy Blood & Tissue Kit (Qiagen). To detect the targeted integration of the HDR cassette in the *AAVS1* locus, two different pair of primers for the 3′ or the 5′ integration junction (5′ TI and 3′ TI, respectively) were used (Supplementary Table S2). PCR was conducted as follows: 2 min at 94°C, 40 cycles of 30 s at 94°C, 30 s at 58°C (5′ TI) and 59°C (3′ TI), 1 min at 72°C and one final step for 5 min at 72°C. The proper target integration amplified a 1195 pb amplicon for the 5′ TI and a 1314 pb fragment for the 3′ TI that were resolved in agarose gel at 2%. For Southern blot analyses, genomic DNA was extracted and digested either with *Bst*XI enzyme or with *Bgl*I (both from New England Biolabs). Matched DNA amounts were separated on 0.8% agarose gel, transferred to a nylon membrane (Hybond XL, GE Healthcare) and probed either with the $^{32}$P-radiolabeled sequence of a fragment of EGFP to detect specific (5.1 kb) and non-specific integrations or with a probe of *AAVS1* gene located outside of the

## The paper explained

### Problem

Gene targeting is becoming a true alternative to conventional gene therapy with integrative gammaretroviral or lentiviral vectors. It is however unknown whether these approaches would be applicable to inherited syndromes like FA, characterized by homology-directed DNA repair (HDR) defects. Additionally, the existence of 16 different FA genes, each of them with multiple mutations potentially accounting for the disease, would imply the necessity of developing individualized targeted gene therapy strategies in FA patients.

### Results

We have demonstrated for the first time an efficient and specific targeting of *FANCA* in the *AAVS1* safe harbor locus of FA-A patients' fibroblasts. This approach allowed us to develop a gene-editing platform applicable to all FA subtypes and FA gene mutations based on the insertion of the therapeutic FA gene in a *safe harbor* locus. Moreover, gene-edited FA-A fibroblasts were reprogrammed to generate disease-free iPSCs, which could be re-differentiated toward the hematopoietic lineage in a process that resulted in the generation of gene-edited, disease-free, hematopoietic progenitor cells.

### Impact

Our data showing that gene targeting is feasible in FA opens the possibility of using similar strategies in different inherited syndromes characterized by defects in HDR and genome instability. The generation of disease-free HSCs through the specific insertion of therapeutic transgenes in a safe harbor locus of non-hematopoietic cell tissues, additionally constitutes an implemented approach to overcome HSC defects characteristic of many DNA repair deficiency syndromes, like Fanconi anemia.

homology arm (in the 3′ region) to detect specific integration in the proper target locus (9.6 kb) and the unmodified *AAVS1* locus (3.3 kb). To detect the radiolabel signal, auto-radiographic films were used (Amershan Hyperfilm ECL, GE Healthcare) and they were exposed in an automatic reveal machine Curix60 (AGFA).

**Supplementary information** for this article is available online: http://embomolmed.embopress.org

## Acknowledgements

The authors would like to thank Prof. Juan C. Izpisua-Belmonte and Dr Guillermo Guenechea for helpful discussions; Laura Cerrato for technical assistance with iPSCs; and Aurora de la Cal for coordination with the FA Network. We are also indebted to the FA patients, families, and clinicians from the FA network. This work was supported by grants to J.A.B. from the European Union (FP7 GA 222878 PERSIST), Spanish Ministry of Economy and Competitiveness (International Cooperation on Stem Cell Research Plan E; Ref PLE 2009/0100; SAF 2009-07164 and SAF 2012-39834), Fondo de Investigaciones Sanitarias, Instituto de Salud Carlos III (RETICS-RD06/0010/0015 and RD12/0019/0023), Dirección General de Investigación de la Comunidad de Madrid (CellCAM; Ref S2010/BMD-2420), and La Fundació Privada La Marató de TV3, 121430/31/32; to J.S. from the Generalitat de Catalunya (SGR0489-2009), the ICREA-Academia program, the Marató de TV3 (464/C/2012), the Spanish Ministry of Science and Innovation (SAF2012-31881), the European Commission (HEALTH-F5-2012-305421), and the European Regional Development FEDER Funds; to L.N. from Telethon (TIGET grant D2), European Union (FP7 GA 222878 PERSIST, ERC Advanced Grant 249845 TARGETINGGENETHERAPY) and the Italian Ministry of Health. The authors also thank the Fundación Marcelino Botín for promoting translational research at the Hematopoietic Innovative Therapies Division of the CIEMAT. CIBERER is an initiative of the Instituto de Salud Carlos III, Spain.

## Author contributions

Contribution: PR, RB, AL, LN, and JAB conceived and designed the experiments. PR, RB, AL, OQ-B, LA, ZG, PG, EA, AV, BD, SN, YT, JPT, and RM conducted experiments. JCS, ES, JS, PDG, and MCH provided reagents, tools, and ideas. PR, RB, AL, LN, and JAB wrote the paper.

## Conflict of interest

P.D.G. and M.C.H. are current or former employees of Sangamo BioSciences, Inc. The rest of the authors declare that they have no conflict of interest.

## For more information

Fanconi Anemia Research Foundation: www.fanconi.org.

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
