## [Review Process File · EMBO Molecular Medicine]

Targeted Gene Therapy and Cell Reprogramming in Fanconi Anemia

P.Río, R.Baños, A.Lombardo, O.Quintana-Bustamante, L.Alvarez, Z.Garate, P.Genovese, E.Almarza, A.Valeri, B.Díez, S.Navarro, Y.Torres, J.P.Trujillo, R.Murillas, J.C.Segovia, E.Samper, J. Surralles, P.D.Gregory, M.C.Holmes, L.Naldini, J.A.Bueren

*Corresponding authors: Juan Bueren, CIEMAT/CIBERER and Instituto de Investigaciones Sanitarias Fundación Jiménez Díaz
Luigi Naldini, San Raffaele Telethon Institute for Gene Therapy, San Raffaele Scientific Institute, and Vita Salute San Raffaele University,*

Review timeline:	Submission date:	09 August 2013
	Editorial Decision:	18 December 2014
	Revision received:	23 March 2014
	Editorial Decision:	08 April 2014
	Revision received:	16 April 2014
	Accepted:	17 April 2014

Transaction Report:

Editor: Céline Carret

1st Editorial Decision

18 December 2014

Thank you for the submission of your manuscript to EMBO Molecular Medicine. We have now heard back from the two referees whom we asked to evaluate your manuscript. Although the referees find the study to be of potential interest, they also raise a number of concerns that need to be addressed in a major revision of the work.

As you will see from their detailed comments, the referees have somehow overlapping issues as both are concerned with novelty/advance, however showing the putative contribution of the FA pathway (or not) in resolving ZFN-induced double strand breaks would make the study more original. In addition, characterization of the genomic stability of the corrected FA fibroblasts should be shown, interpretation on the TERT expression re-evaluated (see comment 3 ref.2), better western blot and statistics provided. Finally, we agree with referee 2 that making the article a short report would focus the study and make it more attractive to our readership.

Given these evaluations, I would like to give you the opportunity to revise your manuscript, with the understanding that the referees' concerns must be fully addressed, experimentally when needed and that acceptance of the manuscript would entail a second round of review.

Please note that it is EMBO Molecular Medicine policy to allow a single round of revision in order

to avoid the delayed publication of research findings. Consequently, acceptance or rejection of the manuscript will depend on the completeness of your responses included in the next version of the manuscript.

I look forward to receiving your revised manuscript.

***** Reviewer's comments *****

Referee #1 (Comments on Novelty/Model System):

The experiments detailed within the manuscript have clearly been performed to a high standard and provide a definitive proof of concept for the use of zinc finger nucleases to facilitate corrective gene editing of Fanconi anemia (FA) patient fibroblasts prior to reprogramming. Although the successful reprogramming of corrected FA patient fibroblasts has been achieved before, this is the first demonstration of ZFN-mediated gene editing in this disease model and this represents a significant step forward in the field.

Referee #1 (Remarks):

Fanconi anemia (FA) is a rare inherited bone marrow failure syndrome, for which the only permanent therapy is bone marrow transplantation from a non-affected donor. One way to overcome the lack of suitable HLA-matched disease-free donors is to genetically correct the patient's own hematopoietic stem cells (HSCs) and then re-infuse these corrected cells back into the patient. Such a gene therapy approach has been attempted in the past but has been severely restricted by both the number and quality of HSC that can be harvested from FA patients. In the current manuscript, Banos et al. propose to overcome this problem by correcting FA patient fibroblasts; reprogramming these cells to pluripotency; then differentiating the pluripotent cells into hematopoietic stem/progenitor cells. Although this approach has been shown to be feasible before, this study is the first to use directed gene editing to correct FA patient fibroblasts as opposed to using retroviral vectors to deliver the complementing FA cDNA. The use of zinc finger nucleases (ZFNs) to facilitate gene editing is of particular interest in the setting of FA, since this technology relies upon the efficient homologous recombination (HR)-mediated repair of a DNA double strand break and FA cells are deficient in mediating some forms of HR DNA repair.

Using a ZFN pair which targets the so called AAVS1 safe harbor in the human genome, coupled with an integration-defective lentiviral vector to act as a donor of a functional FANCA cDNA, the authors were able to correct fibroblasts from four separate FA patients. Notably, corrected fibroblasts demonstrated a prominent growth advantage over their non-corrected counterparts. Corrected fibroblasts could then be reprogrammed using a polycistronic lentiviral vector which over-expressed SOX2, OCT4, KLF4 and cMYC. The delivery of exogenous TERT enhanced the efficiency of both the FA fibroblast gene editing and the reprogramming process. Following excision of the reprogramming cassette, one corrected FA iPS clone could be shown to undergo efficient differentiation into hematopoietic progenitor cells.

This manuscript represents an interesting proof of concept for the use of gene editing to correct FA patient fibroblasts prior to cellular reprogramming. Although this approach is still limited by the lack of suitable protocols with which to generate bona fide transplantable HSC from iPS cells, the fact that the ZFN-mediated introduction of a correcting FA cDNA was successful in the setting of a repair deficient cell is of interest for this disease and a number of other rare inherited disorders that

stem from defects in DNA repair. Of particular note, the selective advantage of gene corrected FA fibroblasts (resulting in a higher than expected frequency of corrected cells) suggests that this approach may be particularly appropriate for this disease model. Nonetheless, there are a number of deficiencies in the manuscript that would result in it being unsuitable for publication in EMBO Mol. Med. in its current format. A detailed critique is provided below.

Major deficiencies

(i) One of the central hypotheses of the manuscript is that ZFN-mediated gene editing will be effective in correcting FA fibroblasts prior to reprogramming, despite the fact that FA fibroblasts are deficient in HR repair. Although the authors are able to demonstrate the generation of corrected FA patient fibroblasts, which can subsequently be used to generate iPS cell lines and then hematopoietic progenitor cells, there is an insufficient characterization of the genomic stability of these cells. That is, the gene editing process is likely to have introduced DNA damage into the patient fibroblasts that cannot be detected via a simple analysis of karyotype. The authors should carry out either exon sequencing or CNV-SNP array analysis on corrected fibroblast/iPS clones compared to the starting patient material.

(ii) The authors perform Sanger sequencing on iPS cells generated from FA patient 52 in order to demonstrate that the patient mutations have not spontaneously reverted during the gene editing/reprogramming process. However, within FA patients, the spontaneous reversion of a defective FA gene in somatic (blood) cells has been documented to occur via the recombination of compound heterozygous loss of function alleles in order to generate one allele that harbors both inactivating mutations and one functional allele. If the authors wish to make this point, then they should demonstrate that the two inactivating mutations remain on separate alleles within the iPS clone (e.g. by sub-cloning the two mRNA species and then sequencing the resulting cDNAs).

(iii) The Western blot demonstrating FANCA expression in iPS clone #26 (Figure 4B) is not at all convincing. The authors should repeat this Western blot with an appropriately loaded gel.

Minor deficiencies

(iv) p8. Alkaline is misspelt.

(v) Figure 3C. Teratoma is misspelt.

Referee #2 (Comments on Novelty/Model System):

Banos et al. have inserted FANCA into the AAVS1 "safe harbour" of FA patient fibroblasts by a ZFN-nuclease-mediated HDR approach and observed proliferative advantage of the transformed cells and noticed an improvement in efficiency of their approach by co-transformation with a hTERT containing LV. Functional analysis of the AASV1/FANCA-containing fibroblasts showed correction of FA phenotype. Subsequently they generated iPS from these FA-corrected fibroblasts and differentiated them into fibroblasts (for genetic characterization) or HSC (for functional characterization).

Overall this paper is interesting and well written. However, some deficiencies are clear and some interpretations are not supported by the available data (see below). In addition, a major premise of the paper is that the FA pathway would limit HDR. However, the authors provide no data substantiating this premise and in the absence of this data, the paper's novelty is compromised, since the technology reported is otherwise been previously reported by the same authors and others.

Major issues:

1. While interesting, the paper describes known techniques and approaches, and the novelty of the data is limited. ZFN nucleases have been used on FA cells before, and the role hTERT

complementation as well. None of the technical approaches are novel, and there is no unexpected finding.

2. The dataset shown is solid but most of it could be shown as supplementary information. iPS characterization assays (teratomas, karyotyping, etc) are probably not main figure findings anymore, as they have become standard basic requirements.

3. The authors utilize Tert expression and show data supporting increased efficiency of reprogramming, but the interpretation of this data are flawed. First, the authors fail to show the FA cells are defective in telomerase activity and also show no data to definitively show that exogenous expression in this model system restores telomerase activity. Second, as even pointed out by the authors themselves, the effect of exogenous expression of Tert is likely non-specific, since it has been reported by other authors to have a similar effect on non-FA cells. Third, As Tert-assisted reprogramming was successful only for a single corrected FA-fibroblast population, it is questionable if this is a generally useful approach in this setting. Although increasing efficiency of gene editing and maybe reprogramming, it represents another genomic insult before correction of the DNA repair machinery (which should at least be discussed, as they claim ZFN-mediated gene editing).

4. The authors suggest that the absence of basal levels of chromosome abnormalities in non-corrected (?-page 7) cells means gene targeting procedures did not generate genetic instability. However, this interpretation may well be wrong, since only cells without significant chromosome instability may be re-programmed. This is particularly relevant given the low numbers of successfully reprogrammed clones in this report.

5. The authors do not discuss at all, why excision of the reprogramming vector has such a strong impact on hematopoietic differentiation, as they showed in Figure 3a and 5a that this vector is fully "silent" anyway. One explanation may be that this simply represents experimental variation. They should have repeated hematopoietic differentiation using the same clone at least once to verify this finding.

6. Several main figures would fit better in the supplementary figures part, as they either show the obvious and well established (like iPS pluripotency) or are redundant (same analyses on fibroblasts, iPS and excised iPS): (1F, 4D (redundant with 2B), 3A, B, C and D and F (iPS pluripotency). Thus this manuscript could be shortened to a 2 page, 2 figure short report. It would clearly make it more attractive a read. If shortened and focused on the main findings (Figure 1B-C-D-E fused with Figure 2A-B to be first figure and Figure 6 to be the second figure) would increase its impact.

7. Much data has no statistical treatment, bar graphs without variance, P values etc.

Minor issues:

1. I would refrain from calling FA a genetic aplasia (abstract)
2. Page 9, "One of these clones, ...was also challenged with DEB and...basal levels of chromosomal aberrations were observed...consistent with a functional FA pathway..." is this as intended? A functional FA pathway should presumably be associated with no reduced levels of the changes, which is what figure shows. The term "basal level" may be confusing.
3. No data on whether the recombinant TERT-expression vector is also silenced during iPS generation nor did the authors test whether this vector is also excised along with the STEMCCA vector.
4. Fig1E: Figure would be more informative, if data from other clones would also be shown, not just #52.
5. The cytogenetic analysis performed on geFA-iPSC (mentioned on page9) should be shown in supplementary figures.
6. No data proving excision of STEMCCA is shown. Did the authors verify excision?

Referee #2 (Remarks):

May be suitable for publication as a short report if major issues above are addressed.

This group is a leader in the field and this study is interesting and important, but it has little novelty and in its current format unfortunately loses some of its impact. In particular, a major premise of the paper is that the FA pathway would limit HDR. However, the authors provide no data substantiating this premise and in the absence of this data, the paper's novelty is compromised, since the technology reported is otherwise been previously reported by the same authors and others. In addition, the data on expression of Tert is compromised in its interpretation as noted in the major comments.

1st Revision - authors' response

23 March 2014

Referee #1 (Comments on Novelty/Model System):

The experiments detailed within the manuscript have clearly been performed to a high standard and provide a definitive proof of concept for the use of zinc finger nucleases to facilitate corrective gene editing of Fanconi anemia (FA) patient fibroblasts prior to reprogramming. Although the successful reprogramming of corrected FA patient fibroblasts has been achieved before, this is the first demonstration of ZFN-mediated gene editing in this disease model and this represents a significant step forward in the field.

The authors would like to thank the reviewer for his/her careful reading of the manuscript and very helpful comments, all of which have been considered in the revised manuscript to further clarify the novelty of our work in the field of FA gene editing and reprogramming.

We appreciate the positive opinion of the reviewer about our work. Specific questions are answered below.

Referee #1 (Remarks):

This manuscript represents an interesting proof of concept for the use of gene editing to correct FA patient fibroblasts prior to cellular reprogramming. Although this approach is still limited by the lack of suitable protocols with which to generate bona fide transplantable HSC from iPS cells, the fact that the ZFN-mediated introduction of a correcting FA cDNA was successful in the setting of a repair deficient cell is of interest for this disease and a number of other rare inherited disorders that stem from defects in DNA repair. Of particular note, the selective advantage of gene corrected FA fibroblasts (resulting in a higher than expected frequency of corrected cells) suggests that this approach may be particularly appropriate for this disease model. Nonetheless, there are a number of deficiencies in the manuscript that would result in it being unsuitable for publication in EMBO Mol. Med. in its current format. A detailed critique is provided below.

The reviewer is absolutely correct in his/her statement that nowadays there are no suitable protocols for the bona fide differentiation of transplantable HSCs from iPSCs. As a proof of concept now we show in one of our excised geFA-iPSCs, that these cells can generate human hematopoietic cells *in vivo* using the teratoma/OP9 approach (see Figure E9).

Major deficiencies

- (i) *One of the central hypotheses of the manuscript is that ZFN-mediated gene editing will be effective in correcting FA fibroblasts prior to reprogramming, despite the fact that FA*

fibroblasts are deficient in HR repair. Although the authors are able to demonstrate the generation of corrected FA patient fibroblasts, which can subsequently be used to generate iPS cell lines and then hematopoietic progenitor cells, there is an insufficient characterization of the genomic stability of these cells. That is, the gene editing process is likely to have introduced DNA damage into the patient fibroblasts that cannot be detected via a simple analysis of karyotype. The authors should carry out either exon sequencing or CNV-SNP array analysis on corrected fibroblast/iPS clones compared to the starting patient material.

Based on the relevant reviewer's observation we have conducted new studies in which CGH arrays have been conducted with FA fibroblasts prior to and after hTERT transduction and gene editing. Additionally, unexcised and excised geiPSCs generated from FA fibroblasts were included in these analyses. As the reviewer will see in the new data, neither TERT-transduction nor gene editing induced detectable genetic alterations in FA cells. In contrast, the reprogramming and/or the iPSC culture and the excision process induced more evident genetic alterations in analyzed cells. This new information has been now included in the manuscript (See results in second par. of page 11; Table I and Figure E7, and discussion in second par. of page 15).

- (ii) *The authors perform Sanger sequencing on iPS cells generated from FA patient 52 in order to demonstrate that the patient mutations have not spontaneously reverted during the gene editing/reprogramming process. However, within FA patients, the spontaneous reversion of a defective FA gene in somatic (blood) cells has been documented to occur via the recombination of compound heterozygous loss of function alleles in order to generate one allele that harbors both inactivating mutations and one functional allele. If the authors wish to make this point, then they should demonstrate that the two inactivating mutations remain on separate alleles within the iPS clone (e.g. by sub-cloning the two mRNA species and then sequencing the resulting cDNAs).*

We agree with the reviewer that recombination of mutated alleles might have occurred in some of our clones, so we have corrected our previous statement in this respect. The problem for performing the study suggested by the reviewer is that one of the pathogenic mutations of patient FA52 is intronic, so it is not possible to sequence it in the cDNA. We also analyzed if there were informative SNPs in the region. Unfortunately, however, we sequenced the entire coding sequence of the *FANCA* gene in this patient's DNA, and there were no informative SNPs to be used as markers of the 710-5T>C at -5 of exon 8. However, supporting our statement that spontaneous reversion should not account for the restored pathway of gene edited FA cells, we have now emphasized that all stable FA iPSCs contained *FANCA* in the *AAVSI* site. Furthermore, the only iPSC clone that had no integration in *AAVSI* locus (iPSC clone 5) could not be maintained for more than 6 passages. This has been clarified in a new paragraph in second par. in page 10.

- (iii) *The Western blot demonstrating FANCA expression in iPS clone #26 (Figure 4B) is not at all convincing. The authors should repeat this Western blot with an appropriately loaded gel.*

A new Western blot in which expression of FANCA is clearly observed in geFA-iPSC clone 26 has been performed and presented in Figure 4.

Minor deficiencies

- (iii) *p8. Alkaline is misspelt.* The misspelling has been corrected.
 (v) *Figure 3C. Teratoma is misspelt.* This misspelling has been also corrected.

Referee #2 (Comments on Novelty/Model System):

The authors would like to thank the reviewer for his/her careful reading of the manuscript and very helpful comments, all of which have been considered in the revised manuscript to further clarify the novelty of our work in the field of FA gene editing and reprogramming.

Overall this paper is interesting and well written. However, some deficiencies are clear and some interpretations are not supported by the available data (see below). In addition, a major premise of the paper is that the FA pathway would limit HDR. However, the authors provide no data substantiating this premise and in the absence of this data, the paper's novelty is compromised, since the technology reported is otherwise been previously reported by the same authors and others.

We agree with the reviewer that this is an important aspect of the work that was not clearly explained in our previous version of the manuscript. We have now clarified in the Introduction that several studies have already formally demonstrated the role of FA proteins in DNA repair, particularly in HDR, as well as in the activity of different nucleases participating in the repair of double strand breaks. We have included two very important reviews published in Nature and Genes & Devel, as well as additional studies demonstrating the relevance of FA proteins in this respect. Since previous studies showed the link between FA and HDR we now clarify in the manuscript that one of the main objectives of our work was to investigate whether or not FA cells, in spite of these deficiencies, could be efficiently targeted with the *FANCA* donor construct (See Introduction; second par. page 3) and not to study the role of FA pathway in HDR, a question that has been clearly demonstrated by others before. To further clarify these very relevant aspects of our work, we have slightly modified the title of our manuscript.

Major issues:

- 1. While interesting, the paper describes known techniques and approaches, and the novelty of the data is limited. ZFN nucleases have been used on FA cells before, and the role hTERT complementation as well. None of the technical approaches are novel, and there is no unexpected finding.*

We hope to have clarified that our manuscript demonstrates for the first time the possibility of performing efficient targeted gene therapy in HDR-deficient cells. Additionally, as far as we know ZFNs have never been used in FA cells.

With respect to the role of hTERT in gene targeting, we have further confirmed the relevance of transducing FA-fibroblasts with hTERT for improving the efficacy of gene targeting (Figure 1C). To the best of our knowledge this is also the first description of such an effect by hTERT.

With respect to the effect of hTERT in cell reprogramming, we agree that this could be an expected finding. We simply used this approach to improve the efficacy of the process. This has been indicated in the discussion (second par. page 14).

- 2. The dataset shown is solid but most of it could be shown as supplementary information. iPS characterization assays (teratomas, karyotyping, etc) are probably not main figure findings anymore, as they have become standard basic requirements.*

As proposed by the reviewer we have moved two figures and several panels from other figures to the expanded view.

3. *The authors utilize Tert expression and show data supporting increased efficiency of reprogramming, but the interpretation of this data are flawed. First, the authors fail to show the FA cells are defective in telomerase activity and also show no data to definitively show that exogenous expression in this model system restores telomerase activity.*

We now demonstrate the lack of telomerase activity in FA fibroblasts and how this activity was restored as a consequence of the transduction with the hTERT-LV. (See results in 2nd par of page 6 and Figure E2).

Second, as even pointed out by the authors themselves, the effect of exogenous expression of Tert is likely non-specific, since it has been reported by other authors to have a similar effect on non-FA cells. Third, As Tert-assisted reprogramming was successful only for a single corrected FA-fibroblast population, it is questionable if this is a generally useful approach in this setting.

We agree that the effect of hTERT in cell reprogramming should not necessarily be specific for FA cells. In fact we used hTERT based on the previous knowledge about the relevance of hTERT in cell reprogramming, so the relevance of hTERT in cell reprogramming was not a specific objective of our study. We have modified our discussion regarding the implications of these experiments, according to the observation of the reviewer. We now mention that our observations regarding the effect of hTERT on cell reprogramming are consistent with data obtained by other authors. (See second par page 14).

Although increasing efficiency of gene editing and maybe reprogramming, it represents another genomic insult before correction of the DNA repair machinery (which should at least be discussed, as they claim ZFN-mediated gene editing).

Although we agree with the reviewer's observation, it is also important to remark that our CGH studies did not show any detectable chromosomal aberration linked to the transduction of FA fibroblasts with hTERT. As proposed by the reviewer, new paragraphs have been introduced in the discussion mentioning this relevant information (See second par page 14).

4. *The authors suggest that the absence of basal levels of chromosome abnormalities in non-corrected (?-page 7) cells means gene targeting procedures did not generate genetic instability. However, this interpretation may well be wrong, since only cells without significant chromosome instability may be re-programmed. This is particularly relevant given the low numbers of successfully reprogrammed clones in this report.*

We have now included a new section in the manuscript related to chromosomal instability studies (see first par. page 11). Our conclusion is that neither transduction with hTERT nor gene-editing induced evident chromosomal abnormalities in FA cells, even though most of these cells had a corrected FA phenotype. In contrast to these observations, more evident chromosomal abnormalities were noted in geFA-iPSCs, strongly suggesting that cell reprogramming and/or the iPSC culture, rather than gene-editing per se, are the most evident processes inducing genetic damage in FA cells (See second par. in page 11 and Figure E7).

5. *The authors do not discuss at all, why excision of the reprogramming vector has such a strong impact on hematopoietic differentiation, as they showed in Figure 3a and 5a that this vector is fully "silent" anyway. One explanation may be that this simply represents experimental variation. They should have repeated hematopoietic differentiation using the same clone at least once to verify this finding.*

As proposed by the reviewer we have performed additional differentiation experiments with two different excised clones, confirming our previous observations. (See Figure E8B). Although the E1a promoter which drives the expression of the reprogramming factors seems to be inactivated in our iPSCs, as the reviewer has noted, the excision of the reprogramming cassette should have further limited the possibilities of reactivation of the reprogramming genes during the hematopoietic differentiation. We have now clarified that our findings are consistent with other studies showing that low levels of expression of the reprogramming genes may interfere with the hematopoietic differentiation (See last par. Page 15).

6. *Several main figures would fit better in the supplementary figures part, as they either show the obvious and well established (like iPS pluripotency) or are redundant (same analyses on fibroblasts, iPS and excised iPS): (1F, 4D (redundant with 2B), 3A, B, C and D and F (iPS pluripotency). Thus this manuscript could be shortened to a 2 page, 2 figure short report. It would clearly make it more attractive a read. If shortened and focused on the main findings (Figure 1B-C-D-E fused with Figure 2A-B to be first figure and Figure 6 to be the second figure) would increase its impact.*

As proposed by the reviewer we have moved two figures and several panels from other figures to the expanded information. Nevertheless, we have had to include additional new relevant data that was asked for by the reviewers. Now the manuscript has 5 figures, 1 table, 9 expanded figures and 1 expanded table. Therefore, in our opinion, the reduction of this manuscript to a Short Report would imply serious difficulties for understanding the meaning and relevance of the results.

7. *Much data has no statistical treatment, bar graphs without variance, P values etc.*

We have included statistical analysis in all figure panels where this was possible (See Figures 2, 4, 5 and E2).

Minor issues:

1. *I would refrain from calling FA a genetic aplasia (abstract):*

As proposed by the reviewer we have now replaced this term with inherited bone marrow failure syndrome.

2. *Page 9, "One of these clones, ..was also challenged with DEB and...basal levels of chromosomal aberrations were observed...consistent with a functional FA pathway..." is this as intended? A functional FA pathway should presumably be associated with no reduced levels of the changes, which is what figure shows. The term "basal level" may be confusing.*

We agree that this paragraph was confusing. We have now included the actual number of chromosomal aberrations in the manuscript and included statistical significance in differences (see first par in page 7 and first par. in page 10).

3. *No data on whether the recombinant TERT-expression vector is also silenced during iPS generation nor did the authors test whether this vector is also excised along with the STEMCCA vector.*

In new analyses we have verified that under optimized excision of iPSCs with the Cre-LV, recombinant TERT is also excised (see last paragraph page 10).

4. *Fig1E: Figure would be more informative, if data from other clones would also be shown, not just #52.*

Since we observed a proliferative advantage of gene edited fibroblasts from four different patients we considered that it was enough to show it once again with one additional sample of TERT-transduced cells (FA-52T fibroblasts).

5. *The cytogenetic analysis performed on geFA-iPSC (mentioned on page9) should be shown in supplementary figures.*

Karyotype and aCGH analyses are now shown in Table I and Figure E7.

6. *No data proving excision of STEMCCA is shown. Did the authors verify excision?*

As proposed by the reviewer, we now provide information regarding the excision of the STEMCCA provirus see last paragraph page 10) and also of the hTERT-LV.

Referee #2 (Remarks):

May be suitable for publication as a short report if major issues above are addressed.

Although we have moved some figures to the supplemental information, we have had to include additional relevant information that was requested by the reviewers. Therefore, in our opinion the reduction of the manuscript to a Short Report would imply serious difficulties for understanding the meaning and relevance of our study.

This group is a leader in the field and this study is interesting and important, but it has little novelty and in its current format unfortunately loses some of its impact. In particular, a major premise of the paper is that the FA pathway would limit HDR. However, the authors provide no data substantiating this premise and in the absence of this data, the paper's novelty is compromised, since the technology reported is otherwise been previously reported by the same authors and others.

With the clarifications already introduced in the manuscript we hope to have clarified that several studies have formally demonstrated the role of FA proteins both in HDR as well as in the activity of different nucleases participating in the repair of double strand breaks. Our work thus demonstrates for the first time that, in spite of such defects, FA cells can be efficiently corrected using a ZFN-mediated gene-editing approach.

In addition, the data on expression of Tert is compromised in its interpretation as noted in the major comments.

As mentioned in point n° 3, new data demonstrates the lack of telomerase activity in FA fibroblasts, something that was restored as a consequence of the transduction with hTERT-LV. (See new data in Figure E2).

Thank you for the submission of your revised manuscript to EMBO Molecular Medicine. We have now received the enclosed reports from the referees that were asked to re-assess it. As you will see the reviewers are now globally supportive and I am pleased to inform you that we will be able to accept your manuscript pending the following final amendments:

- 1) Please provide a brief discussion on point i) highlighted by referee 1
- 2) In light with referee 2, we would suggest a different, shorter and more impactful title. What about something like: "Targeted gene therapy in Fanconi Anemia patients' derived iPSCs"?
- 3) Please reply to the minor comments from referee 2 and include a sharper image for figure 4C as suggested

I look forward to receiving a new revised version of your manuscript as soon as possible.

***** Reviewer's comments *****

Referee #1 (Remarks):

This is a revised version of a manuscript which details the use of zinc finger nucleases to correct fibroblasts of Fanconi anemia (FA) patients by genome editing, prior to the generation of induced pluripotent stem cells (iPSCs) from these cells. These corrected iPSCs could then be differentiated into hematopoietic progenitor cells.

In response to my original critique the authors have added a substantial amount of new data to the manuscript, as detailed below.

(i) Original critique relating to the insufficient characterization of genome stability in gene edited cells:

As requested, the authors have carried out comparative genome hybridization (CGH) analysis to evaluate whether the gene editing process induces genomic instability in FA cells (Table 1 and Figure E7). This data shows that, in one clone, the gene editing procedure does not induce any DNA damage that can be subsequently detected using this technique. It is somewhat unfortunate that the fully reprogrammed iPSC clone acquired detectable DNA damage during the reprogramming process. However, this is well documented to happen during the reprogramming of "normal" somatic cells to pluripotency. The authors should briefly discuss this genome damage in the context of reprogramming non-FA cells, citing appropriate work that already exists.

(ii) Original critique relating to possibility that the patient's original mutations could have spontaneously corrected by recombination of the compound heterozygous FANCA alleles to form one functional copy of this gene (and the author's claims that the endogenous FANCA genes were still inactive):

The author's were not able to perform the experiment that I suggested (sequencing of the two FANCA cDNAs) as one of the mutations is intronic. However, the authors have corrected their original claim and have emphasized that all stable FA iPSCs contained a transgenic FANCA cDNA within the AAVS1 locus (i.e. expression of exogenous FANCA seems to be essential for stable reprogramming).

(iii) Original critique relating to the quality of the western blot demonstrating FANCA expression: This has been repeated and the quality of the new blot is sufficient to make the point that FANCA expression is rescued in gene edited cells.

The authors have also corrected the typo's that I highlighted under minor deficiencies.

Taken together, I would judge that the author's have adequately addressed my original comments

and, assuming that they can now provide a brief statement relating to reprogramming-induced genome instability using non-FA somatic cells (point (i) above), then I would judge that this work is now suitable for publication in EMBO Mol. Med.

Referee #2 (Remarks):

1. Overall improved.
2. Title is not improved; actually it is less than optimal.
3. The data in this manuscript are still most c/w a short report as much of the data presented in figures are repetitive.
4. The discussion could be improved with a paragraph on the limitations of the current approach, particularly given the presence of cytogenetic abnormalities in the generated reprogrammed clones.

Minor comments

1. In excised clone 16.1 why was 0.4 copies /cell of the STEMCCA vector detected?
2. On page 8, there are two statements which might be improved vis a vis accuracy. "Consistent with previous observations (Raya et al)..." and "no stable iPSC lines could be generate...most probably due to pro-senescence nature..." ignores subsequent publication by Muller et al. (cited regularly elsewhere in the paper) that shows reprogramming can occur in FA-deficient cells albeit with less efficiency.
3. Figure 4C is technically inferior; it is difficult to distinguish DNA damage foci in these photomicrographs.

2nd Revision - authors' response

16 April 2014

Many thanks for your positive opinion of our manuscript with Reference EMM-2013-03374).

As suggested we have included the proposed amendments (marked in red in the new version of the manuscript):

- 1) A brief discussion proposed by referee 1 (point i) has been included in page 15 par 2.
- 2) The title of the manuscript has been replaced by a shorter and more impactful title. It is similar to the one that you proposed, although we preferred not to say that we have targeted FA iPSCs, since the targeting process was performed prior to cell reprogramming.
- 3) Minor comments of referee 2 have been included:
 - 3.1. Instead of 0.4 copies we have been more precise now, and included the number of 0.35 ± 0.10 copies/cell. This is close to reprogramming-free cells based on Q-PCR analyses (Charrier et al, 2010), but we cannot exclude the possibility of a minor contamination with not excised cells.
 - 3.2. As proposed by the reviewer, we have included the reference of Muller et al and also of Yung et al in pag 15 par 1, mentioning that reprogramming of FA cells is possible, albeit at a very low efficiency.
 - 3.3. We have improved the quality of Figure 4C.

We very much hope that you would be satisfied with these amendments, and therefore that the manuscript could be ready for publication in EMBO Molecular Medicine.